# HIF-1α: A Key Factor Mediating Tumor Cells from Digestive System to Evade NK Cell Killing via Activating Metalloproteinases to Hydrolyze MICA/B

**DOI:** 10.3390/biom15060899

**Published:** 2025-06-19

**Authors:** Quan Zhu, Shuyi Tang, Ting Huang, Chunjing Chen, Biyuan Liu, Chuyu Xiao, Liugu Chen, Wang Wang, Fangguo Lu

**Affiliations:** 1Department of Immunology, School of Medicine, Hunan University of Chinese Medicine, Changsha 410208, China; 004931@hnucm.edu.cn (Q.Z.); 202308040323@stu.hnucm.edu.cn (S.T.); 004205@hnucm.edu.cn (B.L.); 202408020133@stu.hnucm.edu.cn (C.X.); ad@stu.hnucm.edu.cn (L.C.); wang18038882005@stu.hnucm.edu.cn (W.W.); 2Department of Pathology, School of Medicine, Hunan University of Chinese Medicine, Changsha 410208, China; 004929@hnucm.edu.cn; 3Department of Pathogenic Biology, School of Medicine, Hunan University of Chinese Medicine, Changsha 410208, China; 004787@hnucm.edu.cn

**Keywords:** immune escape, hypoxia-inducible factor-1α, metalloproteinase, NK cells, NKG2D, MICA

## Abstract

Malignant tumors of the digestive system are widespread and pose a serious threat to humans. Immune escape is an important factor promoting the deterioration of malignant tumors in the digestive system. Natural killer cells (NK cells) are key members of the anti-tumor and immune surveillance system, mainly exerting cytotoxic effects by binding to the activating receptor natural killer cell group 2D (NKG2D) on their cell surface with the corresponding ligands (major histocompatibility complex class I chain-related protein A/B, MICA/B) on the surface of tumor cells. Malignant tumors of epithelial origin usually highly express NKG2D ligands such as MICA, which can attract NK cells to kill tumor cells and also serve as an important basis for NK cell-based immunotherapy. Tumor cells highly express hypoxia-inducible factor-1α (HIF-1α), which promotes the expression of matrix metalloproteinases (MMPs) and a disintegrin and metalloproteinases (ADAMs). These metalloproteinases hydrolyze MICA and other ligands on the surface of tumor cells to generate soluble molecules. These soluble ligands, when binding to NKG2D, cannot activate NK cells and also block the binding of NKG2D to MICA on the surface of tumor cells, enabling tumor cells to evade the killing effect of NK cells. Almost all organs in the digestive system originate from epithelial tissue, so the soluble ligands generated by the HIF-1α/MMPs or HIF-1α/ADAMs signaling pathways play a crucial role in evading NK cell killing. A comprehensive understanding of this immune escape process is helpful for a deeper understanding of the molecular mechanism of NK cell anti-tumor activity. This article reviews the molecular mechanisms of common digestive system malignancies evading NK cell killing, providing new insights into the mechanism of tumor immune escape.

## 1. Introduction

Malignant tumors of the digestive system mainly include gastric cancer, colorectal cancer, liver cancer, pancreatic cancer, gallbladder cancer, and cholangiocarcinoma. Most of them originate from epithelial cells and are characterized by insidious onset, high malignancy, and poor treatment outcomes, seriously threatening the lives of patients. Malignant tumors of the digestive system have a low response rate to traditional treatments, and immunotherapy is a promising new approach. However, these tumors have complex immune escape mechanisms, which are major obstacles to their immunotherapy [1,2,3]. Immune cell therapy, mainly based on natural killer (NK) cells, has shown superior effects in basic research [4,5], but the immune escape of tumor cells against NK cells hinders the cytotoxic effect of NK cells on tumor cells [6,7]. Therefore, clarifying the immunological principles of the immune escape of digestive system malignant tumors from being killed by NK cells is of great importance for the future development of NK cell therapy for tumors.

The NK cell is one of the most important innate lymphoid cells in the body and is a key element of immune surveillance. NK cells are in an inhibited state under physiological conditions to avoid abnormal activation, causing pathological damage to tissues and organs. After cells undergo tumor transformation [8] or viral infection [9], they can upregulate a series of chemotactic or activating factors and recruit and activate NK cells to play cytotoxic effects. Their activation ability is closely related to the occurrence, development, and clinical outcome of various tumors [10,11].

The activation or inhibition of NK cells depends on the signal balance of a series of activating or inhibitory receptors expressed on the cell surface. NKG2D is the most important activating receptor of NK cells [12]. NKG2D is encoded by the human *KLRK1* gene, and it is a C-type lectin-like receptor expressed on the surface of NK cells [13], CD8^+^ T cells [14], and γδ T cells [15]. There are many types of ligands for NKG2D. In humans, major histocompatibility complex class I-like proteins A/B (MICA, MICB) and UL16-binding proteins 1-6 (ULBP1-6) are common ligands for NKG2D. These ligands are almost not expressed on the surface of normal tissue cells in humans but are only lowly expressed on the surface of epithelial cells and vascular endothelial cells [16,17]. When normal cells (especially those derived from epithelial tissues) undergo tumor transformation, the expression levels of MICA and other ligands increase significantly, attracting NK cells and binding to the NKG2D receptor [8]. NK cells are activated through a series of signal transductions and then perform the function of lysing target cells. The type of cells from the gastrointestinal tract, liver, gallbladder, and pancreas are almost all epithelial cells, and their expression of MICA and other ligands increases significantly after malignant transformation [18,19,20,21,22].

A high expression of MICA or other ligands on the surface of tumor cells can be regarded as an immune checkpoint. After binding to NKG2D, they can effectively activate NK cells and CD8^+^ T cells to play an anti-tumor role [23,24]. However, tumor cells have also evolved a unique immune escape mechanism under pressure selection, that is, downregulating the expression of NKG2D ligands on the cell surface or generating soluble ligands. These soluble ligands can competitively bind to NKG2D, preventing the activation of the downstream signaling pathway of NKG2D and the activation of NK cells [25]. Moreover, soluble ligands occupy the binding domain of NKG2D on the surface of tumor cells, making it impossible for NK cells to kill tumor cells. This is an important reason for tumor escape from immune killing and the worsening of the disease [26]. Therefore, a comprehensive understanding of the regulatory mechanism of NKG2D and its ligands plays a very crucial role in the body’s anti-tumor immunity and the prognosis of patients and is also the theoretical basis for the treatment of tumors using the MICA-NKG2D signaling axis in the field of tumor immunology.

Many solid tumor cells are in a highly dense hypoxic microenvironment. To survive and proliferate in this environment, tumor cells highly express hypoxia-inducible factor-1α (HIF-1α), which can trigger glycolysis in tumor cells and provide energy through anaerobic respiration [27]. Even when oxygen is abundant, tumor cells still maintain high levels of HIF-1α for survival [28]. HIF-1α has been reported to be related to the production of soluble NKG2D ligand molecules or to reduce the expression of NKG2D ligands on tumor cell surfaces and is a key factor in mediating tumor immune escape mechanisms against NK cells and others [29,30,31]. According to most studies, HIF-1α regulates the production of soluble NKG2D ligands by activating multiple downstream metalloproteinases, including matrix metalloproteinases (MMPs) and a disintegrin and metalloproteinases (ADAMs) [32,33,34]. These metalloproteinases directly hydrolyze ligands such as MICA on the cell membrane surface in the form of exosomes to produce soluble ligands. After soluble ligands bind to NKG2D, they induce the internalization or degradation of NKG2D receptors on the surface of NK cells and CD8^+^ T cells, inhibiting the cytotoxicity of these immune cells and thereby mediating tumor immune escape [35,36]. Hence, the HIF-1α/MMPs or HIF-1α/ADAMs signaling pathways can be regarded as the key elements for immune escape against NK cells in digestive system tumors.

We reviewed the HIF-1α/MMP or HIF-1α/ADAM signaling pathways and immune escape mechanisms in common digestive system tumors. The abnormally high expression of these two signaling pathways in tumors leads to the massive shedding of ligands such as MICA, which is conducive to tumor progression. Therefore, the regulation of the HIF-1α/MMP and HIF-1α/ADAM signaling pathways is of great importance for the clinical prognosis of tumor patients. MICA is mainly expressed on the surface of tumor cells derived from epithelial cells, and digestive system malignancies are the main types of tumors derived from epithelial cells. Therefore, we describe in detail the mechanism by which the HIF-1α/MMP and HIF-1α/ADAM signaling pathways regulate the generation of soluble ligands to mediate immune escape in digestive system malignancies, providing a theoretical reference for in-depth understanding of the immune escape mechanisms of digestive system malignancies and clinical immunotherapy targeting these tumors.

## 2. MICA/B and NKG2D Are Imbalanced in Expression in Tumor Tissues of the Digestive System

The expression of MICA and MICB in common digestive system malignant cancers was analyzed using TIMER software (https://cistrome.shinyapps.io/timer/, accessed on 10 April 2025). MICA and MICB expression levels in digestive system malignant tumors are generally higher than in relatively normal tissues. The expression levels of MICA and MICB in cholangiocarcinoma (CHOL), colorectal cancer (COAD), hepatocellular carcinoma (LIHC), and gastric cancer (STAD) were significantly higher than those in the corresponding normal tissues, while there was no statistically significant difference in the expression level in pancreatic cancer (PAAD) compared with normal pancreatic tissues (Figure 1A,B).

NKG2D is the receptor of MICA and MICB. However, the expression of NKG2D in various tumors is significantly lower than that in the corresponding normal tissues. In malignant tumors of the digestive system, the expression level of NKG2D in colorectal cancer (COAD), hepatocellular carcinoma (LIHC), and rectal adenocarcinoma (READ) was significantly lower than that in the corresponding normal tissues. There was no statistically significant difference in the expression level of NKG2D in gastric cancer (STAD), cholangiocarcinoma (CHOL), and pancreatic cancer (PAAD) compared with the corresponding normal tissues (Figure 1C).

MICA and MICB are not expressed or expressed at a low level in normal tissue cells but are highly expressed on the surface of various tumor cells. As important ligand molecules of the NK cell-activating receptor NKG2D, MICA and MICB on the surface of tumor cells activate the anti-tumor effect of NK cells after binding to NKG2D. The expression level of NKG2D in tumor tissues is usually lower than that in the corresponding normal tissues, suggesting that the tumor may inhibit the infiltration of NK cells in some ways and cause immune escape. In digestive system tumors, various tumors highly express MICA or MICB, but the expression level of NKG2D is lower than that in normal tissues. Data from the TIMER software (ttps://cistrome.shinyapps.io/timer/, accessed on 10 April 2025).

## 3. Abnormal Activation of Metalloproteinases Leads to the Shedding of MICA/B on the Tumor Cell Surface to Form Soluble Ligands

### 3.1. Metalloproteinases Are Generally Highly Expressed in Digestive System Tumors

Matrix metalloproteinases (MMPs) are a large class of zinc-dependent endopeptidases that require the assistance of metal ions such as Ca^2+^ or Zn^2+^ to exert biological functions. Physiologically, MMPs are of great significance for the remodeling of tissue structure and maintaining the integrity of cell membranes [37]. Pathologically, MMPs can be involved in the occurrence and development of various diseases, such as cancers [38], inflammation [39], infection [40], and cardiovascular diseases [41]. At present, 26 members of the MMP family have been identified, with the numbers being MMP1-26. According to the substrates and similarity degree of amino acid sequences of MMPs, the members of the MMP family were divided into five subcategories: collagenases (MMP1, MMP8, MMP13), gelatinases (MMP2, MMP9), hemolytic enzymes (MMP3, MMP10, MMP11), membrane-type MMPs (MT MMPs), and other MMPs. Among them, MMP2 and MMP9 are also known as type IV collagenases, which are the most commonly expressed MMPs in tumor cells and are highly expressed in digestive system tumors (Figure 2A,B). MMP9 is glycosylated based on MMP2. Most MMPs consist of a precursor peptide of approximately 80 amino acids, a catalytic domain of 170 amino acids, a variable-length linking peptide, and a heme domain of about 200 amino acids. Physiologically, MMPs remain at trace levels in the body; multiple cytokines (such as IL-1, IL-4, IL-6, and TGF-β) can regulate the expression levels of MMPs to maintain the stability of cell membranes and normal function [42,43,44,45,46]. Highly expressed MMP2 and MMP9 can widely degrade various proteins on tumor cell membranes and have become important targets in the field of cancer treatment.

A disintegrin and metalloproteinases (ADAMs) are another type of metalloproteinases that are widely involved in the occurrence and development of various cancers. There are more than 30 family members in ADAMs, and the Zn^2+^ binding domain is also included. Most ADAMs have a cysteine-rich domain and an EGF-like domain adjacent to the transmembrane region, followed by an intracellular region whose length and sequence vary greatly among different ADAM family members. Normally, ADAMs can act on proteins on the cell membrane and hydrolyze them, turning them into soluble forms that enter the blood or tissue fluid [46]. In the field of oncology, ADAM10 is one of the most common and important metalloproteinases, which can be highly expressed in various tumors and hydrolyze membrane proteins on the cell membrane surface (Figure 2C, data from the GEPIA 2.0 software, http://gepia2.cancer-pku.cn/, accessed on 10 April 2025).

MMP2 is highly expressed in pancreatic cancer (PAAD), and there is no statistically significant difference in its expression in other digestive system tumors compared with the corresponding normal tissue (Figure 2A). The expression of MMP9 in cholangiocarcinoma (CHOL) showed no statistically significant difference from that in normal bile duct tissue but was significantly increased in other digestive system tumors (Figure 2B). The expression of ADAM10 in cholangiocarcinoma (CHOL) and liver cancer (LIHC) showed no statistically significant difference from that in normal bile ducts and livers but was significantly increased in other digestive system tumors (Figure 2C). However, it should be noted that there are differences between bioinformatics analysis and literature reports. The expression of these three metalloproteinases has been reported to have significantly elevated in almost all digestive system tumors. Data from the GEPIA 2.0 software, http://gepia2.cancer-pku.cn/, accessed on 10 April 2025).

### 3.2. Metalloproteinases Promote the Generation of sMICA and sMICB to Mediate Tumor Immune Escape Against NK Cells

Malignant tumors derived from the digestive system, such as gastric cancer, colorectal cancer, and liver cancer, all have high levels of MICA or MICB expression, as well as high levels of MMPs or ADAMs [47,48,49]. These metalloproteinases hydrolyze MICA or MICB to generate soluble MICA (sMICA) or soluble MICB (sMICB). Both sMICA and sMICB can bind to the activating receptor NKG2D on the surface of NK cells. However, after binding, they cannot activate the downstream signaling pathway of NKG2D, and they cannot activate NK cells to exert the effect of killing tumor cells; these soluble ligands occupy the binding domain of NKG2D and MICA/B, which are expressed on tumor cells [50]. It causes tumor cells to evade the killing effect of NK cells, CD8^+^T cells, and γδT cells, then promoting tumor progression.

Several studies have reported in detail the process by which metalloproteinases promote the generation of sMICA or sMICB and mediate tumor immune escape. The occurrence, development, and immune escape mechanism of liver cancer are closely related to the production of sMICA/B mediated by MMP2 and MMP9. Cheung et al. found that the expression level of granulin epithelin precursor (GEP) was significantly higher in liver cancer cells; the expression of GEP is negatively correlated with the expression of MICA on the surface of hepatoma cells. Highly expressed GEP can promote the expression of more inhibitory molecules, such as HLA-E, in hepatoma cells and induce the generation of sMICA to reduce the killing activity of NK cells. Mechanistically, MMP2 and MMP9 promote sMICA generation by regulating GEP. After blocking GEP with antibodies, the killing of hepatoma cells by NK cells can be enhanced [51]. The process by which MMP2 regulates the generation of sMICA is also affected by CD133 in hepatoma cells. Kohga et al. found that silencing CD133 could significantly reduce the expression of MMP2 in hepatoma cells, thereby increasing the expression of MICA on the cell membrane surface and reducing the production of sMICA, suggesting that metalloproteinases are directly involved in the immune escape against NK cells in liver cancer [52].

In addition to regulating the expression of MICA in hepatoma cells, metalloproteinases can also regulate the activity of immune cells in the tumor microenvironment and thereby affect the progression of liver cancer. Wu et al. performed single-cell sequencing on patients with advanced liver cancer and found that MMP9 was highly expressed in macrophages in the tumor microenvironment and induced M2 polarization in macrophages. After patients were treated with chemotherapy, the DNA of hepatoma cells was damaged, and interferon regulatory factor 1 (IRF1) was activated, stimulating M2 macrophages to secrete a large amount of MMP9 in the form of exosomes to hydrolyze MICA on the surface of hepatoma cells. Subsequently, sMICA promoted the tumor cells to evade the killing by NK cells [53].

ADAM10 also plays an important role in the immune escape of liver cancer. Otoyama et al. pointed out that retinoids can effectively inhibit ADAM10 and reduce the generation of sMICA by liver cancer cells [54]. Some other drugs also have similar immunomodulatory effects. Ke et al. reported that the expression of MICA on the surface of the hepatoma cell line HepG2 treated with 5-FU was significantly increased. 5-FU prevented MICA shedding from the cell membrane of HepG2. At the same time, Clostridium difficile oocyte polysaccharide (SEP) was used to activate NKG2D and its downstream DAP10/PI3K/Erk signaling pathway with 5-FU, enhancing the killing effect of NK cells on HepG2 cells. However, the overexpression of ADAM10 in HepG2 cells can counteract this synergistic effect [55]. Goto et al. reported that the anti-alcohol drug disulfiram inhibits ADAM10 in hepatoma cells, significantly increases the MICA expression on the cell surface, and reduces the production of sMICA. Disulfiram has no obvious toxic side effects on normal liver cells and is a new potential way to enhance the body’s anti-liver cancer immunity. It also indicates that the generation of sMICA induced by ADAM10 is an important reason for hepatoma cells to evade the killing by NK cells [56]. The role of ADAM10 in cholangiocarcinoma is the same as that in hepatocellular carcinoma. Oliviero et al. reported that the expression levels of MICA and MICB on the surface of cholangiocarcinoma cells were enhanced with the increase in tumor differentiation degree. sMICA in the peripheral blood of cholangiocarcinoma patients is elevated and consistent with the high expression levels of ADAM10 and ADAM17. Regulating the activities of ADAM10 and ADAM17 to inhibit the shedding of MICA/B on cancer cells can significantly promote the degranulation of peripheral, liver, and tumor-infiltrating NK cells and the production of IFN-γ, enhancing the anti-tumor activity of NK cells [8].

Metalloproteinase-mediated tumor cell evasion of the NK cell-killing effect has also been reported in other digestive system tumors. When Shiraishi et al. conducted research on gastric cancer cell lines and gastric cancer specimens, they found that the expression of multiple NKG2D ligands was downregulated, accompanied by the upregulation of MMP9 expression. The expression levels of the two were negatively correlated, the concentrations of sMICA and other soluble ligands were significantly decreased, and the killing activity of NK cells was enhanced after the inhibition of the expression of MMP9 [57]. This indicates that the upregulation of MMP9 activity can mediate the immune escape against NK cells by increasing the levels of soluble ligands such as sMICA in gastric cancer cells. Ou et al. pointed out that the concentration of sMICA secreted by Panc-1 cells, a kind of pancreatic cancer cell line, was significantly increased under hypoxic conditions, and miR-153 could reduce its expression by targeting HIF-1α and ADAM10. The NKG2D receptor of NK cells in pancreatic cancer patients with high expression of HIF-1α was internalized, then lost the ability to bind to NKG2D ligands on the surface of tumor cells, and NK cells could not exhibit cytotoxicity to pancreatic cancer cells [58].

The above studies indicate that metalloproteinases mediate the generation of sMICA and sMICB in malignant tumor cells of the digestive system. High expression of metalloproteinases means that tumor cells produce high levels of sMICA and sMICB, leading to the worsening and progression of cancers.

## 4. Soluble MICA and MICB Impair NKG2D Signaling, Which Is a Key Mechanism for Tumor Evasion of NK Cell-Mediated Cytotoxicity

The purpose of tumor cells expressing MICA or MICB is to activate NK cells, γδT cells, or CD8^+^T cells to kill tumor cells and promote anti-tumor immunity. However, the high expression of metalloproteinases by tumor cells is a key mechanism for evading the cytotoxicity of NK cells and other immune cells.

Luo et al. found that when the concentration of serum sMICA of HCC patients was below 400 pg/mL, the average survival time was significantly longer than that of patients with sMICA levels above 400 pg/mL. They found that sMICA hindered the activation of NKG2D signaling in NK cells, reduced the generation of cytokines such as IFN-γ, TNF-α, and IL-8 by NK cells, and decreased the expression of CD107a, a specific marker of NK cell activation, thereby impairing the anti-tumor immune response of the body [59]. Furthermore, Mantovani et al. found that high levels of sMICA could significantly inhibit the function of NK cells in HCC patients, manifested as a reduction in CD107a and NKG2D; the secretion of IFN-γ was also decreased. The expression of CD107a and IFN-γ by NK cells in HCC patients had been increased after treatment with IL-15, but these NK cells in a high-concentration sMICA environment still could not be fully activated, indicating that sMICA impairs NK cell function [60]. Kohga et al. showed that the serum sMICA concentration in HCC patients was significantly reduced after treatment with transcatheter arterial embolization (TAE), and the expression of NKG2D on NK cells and CD8^+^T cells was subsequently upregulated in those patients. This suggests that timely intervention in the sMICA of HCC patients can delay disease progression and restore the cytotoxic function of NK cells and CD8^+^T cells. However, they also found that the serum sMICB concentration in HCC patients could not be reduced by TAE; the reason and molecular mechanism for this remain to be further explored. NK cells play an important anti-gastric cancer immune system role [61]. Zhao et al. reported that the concentrations of serum sMICA and sMICB were elevated in gastric cancer patients, and the cytotoxicity of NK cells to gastric cancer cells was limited, leading to the evasion of NK cell-mediated cytotoxicity by gastric cancer cells. High concentrations of sMICA and sMICB inhibited the activation of NK cells, suggesting that the activation of NK cells has important clinical significance in the outcome of gastric cancer [62]. To promote NK cell activation, Chen et al. designed a bifunctional protein, NKG2D-IL-15, which binds to MICA and other NKG2D ligands on the surface of gastric cancer cells through the extracellular domain of NKG2D and then binds to IL-15 receptor on the surface of NK cells through IL-15, bringing NK cells and gastric cancer cells closer together and promoting the cytotoxicity of NK cells, thereby enhancing the anti-tumor immunity of patients [63].

Pancreatic cancer is a highly malignant digestive system tumor that progresses rapidly and is difficult to detect in the early stage. Immune escape plays an important role in promoting the rapid worsening of pancreatic cancer. Duan et al. found that the serum sMICA was elevated in pancreatic cancer patients, and the sMICA level in patients with low differentiation was significantly higher than that in patients with high differentiation. The increase in sMICA was associated with the downregulation of NKG2D expression and the impairment of NK cell activity. After tumor resection, the serum sMICA level significantly decreased, and NKG2D expression increased. The change in sMICA level was negatively correlated with NKG2D expression. They found that the survival time of pancreatic cancer patients with a concentration of serum sMICA below 290 pg/mL was significantly longer than that of other pancreatic cancer patients with a concentration of serum sMICA above 290 pg/mL, and also found that pancreatic cancer patients with high expression of MICA on tumor cells had longer survival times, suggesting that MICA could be a potential therapeutic target for pancreatic cancer [64].

Similarly to sMICA, sMICB is also an important factor in the progression of pancreatic cancer and the evasion of NK cell-mediated cytotoxicity by pancreatic cancer cells. Morisaki et al. found that gemcitabine could upregulate the expression of MICA and MICB on pancreatic cancer cells and induce NK cell activation. However, the synergistic effect of gemcitabine on NK cells was blocked when adding sMICB, as sMICB blocked the NKG2D receptor to limit the activation of NK cells. This phenomenon was verified in the treatment of advanced pancreatic cancer patients with gemcitabine combined with an NK cell infusion, which is a potentially valuable combined treatment method in clinical practice [65].

MICA and MICB are also expressed on the surface of colorectal cancer cells derived from intestinal epithelium cells. Zhang et al. found that MICA and MICB were highly expressed in colorectal cancer tissues, and the levels of serum sMICA and sMICB from colorectal cancer patients were also higher than those of the healthy population. sMICA and sMICB significantly downregulated the expression of NKG2D on the surface of NK cells, directly inhibiting NK cell activation and promoting the metastasis of colorectal cancer. They proposed that immunotherapy based on NK cells after removing sMICA and sMICB from the patient’s blood might be a potential new method for treating colorectal cancer [66].

## 5. The Key Factor for Digestive System Tumors to Evade Killing by NK Cells Is That the HIF-1α Promotes the Expression of METALLOPROTEINASE

### 5.1. HIF-1α Is Highly Expressed in Tumor Tissues and Negatively Correlated with the Survival Time of Patients

The expression of HIF-1α in digestive system tumors was analyzed using GEPIA 2.0 software. The expression of HIF-1α in pancreatic cancer (PAAD) and gastric cancer (STAD) was significantly higher than that in the corresponding normal tissues, while there was no significant difference in the expression of HIF-1α in other digestive system tumors compared with the corresponding normal tissues.

However, many studies have shown that although HIF-1α expression in some digestive system tumors shows no statistical differences in the GEPIA database; in fact, HIF-1α has been reported to be highly expressed in almost all malignant tumors of the digestive system and negatively correlated with survival time, which is far from the phenomenon described in the bioinformatics database (Table 1).

### 5.2. Activation of Multiple Metalloproteinases by HIF-1α Is an Important Cause of Soluble MICA/B Generation

Almost all solid tumors have a highly hypoxic microenvironment, and tumor cells highly express HIF-1α to maintain survival through glycolysis [73,74]. HIF-1α is closely related to tumor progression and can induce the activation of multiple signaling pathways related to cell growth, as well as the activation of multiple metalloproteinases to hydrolyze NKG2D ligands such as MICA on tumor cell membranes, promoting the generation of sMICA or sMICB and immune escape against NK cells.

Vascular mimicry (VM) refers to the ability of highly invasive tumor cells to connect with each other to form new vascular networks, which can lead to tumor metastasis and is associated with poor prognosis in patients. Zong et al. found that colon cancer cell lines HCT-116 and LoVo can form VM networks, and the highly hypoxic microenvironment causes colon cancer cells to produce a large amount of reactive oxygen species and increase the expression of MMP2. The traditional Chinese medicine compound Baizhu Huangqi Mixture (AAM) has the effect of inhibiting VM formation. AAM inhibits VM formation in colon cancer cells through the inhibition of the expression of HIF-1α, thereby reducing the pro-angiogenic effect of MMP2. In a mouse lung metastasis model, AAM can reduce the lung metastasis ability of colon cancer cells by downregulating the activation of the HIF-1α/MMP2 signaling pathway [75]. Rassouli et al. found that sunitinib treatment of LoVo cells significantly inhibited the migration and invasion ability and effectively reduced the expression levels of HIF-1α, MMP2, and MMP9 in LoVo cells [76]. Zeng et al. reported that the expression of MMP2, MMP9, and VEGF was significantly reduced after treatment with an HIF-1α inhibitor, and the VM ability of tumor cells was also significantly limited, suggesting that the HIF-1α/MMP2 or HIF-1α/MMP9 signaling pathway is involved in the proliferation and metastasis of colon cancer, fully demonstrating that HIF-1α exerts a biological regulatory function on colon cancer cells through MMP2 or MMP9 [77].

Gallbladder cancer is a rare malignant tumor in the digestive system with limited treatment options and a high degree of malignancy, usually occurring in people with long-term cholelithiasis. Immune escape is an important reason for the progression of gallbladder cancer. Kawamoto found that the primary gallbladder cancer specimens had high expression of myosin-related kinase B (TrkB), and the overall survival rate of patients with high expression of TrkB in the tumor invasion front was lower than that of patients with low expression of TrkB. TrkB can promote the expression of MMP2, VEGF-A, VEGF-C, and VEGF-D, promoting the formation of new blood vessels in gallbladder cancer cells, and the cause of this is that TrkB promotes the high expression of HIF-1α in gallbladder cancer tissues, leading to multiple malignant phenotypes in gallbladder cancer. Since TrkB is highly expressed in gallbladder cancer patients, TrkB or HIF-1α may be promising therapeutic targets. Moreover, HIF-1α can also regulate the expression of MMP9 in gallbladder cancer [78]. Chen et al. proposed that oridonin (a biologically active diterpene compound isolated from Rabdosia rubescens) has potential anti-gallbladder cancer effects. Oridonin can significantly reduce the expression of HIF-1α and MMP9 in GBC-SD cells, blocking the HIF-1α/MMP9 signaling pathway in cells, and this has been verified in mouse models [79]. From these experiments, it is found that the abnormal activation of the HIF-1α/MMPs signaling pathway is an important cause of the proliferation and metastasis of gallbladder cancer.

HIF-1α acts as a significant factor in the occurrence and development of gastric cancer. Implantation metastasis is one of the main mechanisms of metastasis and is also the main cause of the high mortality rate of gastric cancer. Huang et al. pointed out that the expression of HIF-1α can promote the epithelial–mesenchymal transition (EMT) of gastric cancer cells, while dextran sulfate (DS) can inhibit HIF-1α and reduce the expression of other molecules closely related to HIF-1α, including MMP2 and TGF-β. Treating gastric cancer cells with DS or silencing HIF-1α can significantly inhibit the volume and number of metastatic tumors in a mouse model of gastric cancer and suppress TGF-β-mediated EMT [80]. Han and Tsai found that inhibiting the expression of VEGF in gastric cancer cells can reduce the expression of MMP2 and MMP9. Because HIF-1α is an upstream regulatory factor of VEGF, the reduced activity of the HIF-1α/VEGF signaling pathway indicates a better prognosis for gastric cancer patients. In addition, IL-32 also has the effect of inducing the expression of MMP2 and MMP9, and the inactivation of HIF-1α can reduce the expression of IL-32, leading to a decrease in the expression of MMP2 and MMP9, inhibiting the proliferation and metastasis of gastric cancer, and providing a more comprehensive molecular mechanism to confirm the role of the HIF-1α/MMP2 and HIF-1α/MMP9 signaling pathways in the occurrence and development of gastric cancer [81,82]. In addition, inhibiting the upstream regulatory factors of HIF-1α is also helpful for the treatment of gastric cancer. Osteopontin (OPN) is a secreted integrin-binding matrix phosphorylated glycoprotein. OPN has been proven to promote the progression and metastasis of malignant tumors and has prognostic value in several cancers, including gastric cancer. It has been reported that OPN is correlated with HIF-1α or MMP9. Overexpression of OPN in the gastric cancer cell line SGC7901 can significantly promote cell invasion and metastasis. Mechanistically, OPN can bind to the α_v_β_3_ integrin protein of gastric cancer cells, activate the PI3K/Akt signaling pathway, and then upregulate the expression of HIF-1α. Subsequently, the expression of MMP2 and MMP9 in SGC7901 cells is increased. Targeting OPN can block the PI3K/Akt/HIF-1α signaling pathway and is a potentially valuable approach for targeted therapy of gastric cancer [83]. Han et al. discovered a small molecule compound that can inhibit the expression of CD147 in gastric cancer cells, reduce the activation of the MAPK signaling pathway mediated by CD147, and arrest the cell cycle of gastric cancer cells at G2/M, leading to the inactivation of the PI3K/AKT/mTOR signaling pathway and subsequently inhibiting the downstream HIF-1α/VEGF signaling pathway and HIF-1α/MMP signaling pathway, thereby delaying the progression of gastric cancer [84].

Liver cancer is a common malignant tumor of the digestive system. HIF-1α acts as a transcription factor to regulate the expression of downstream target genes. You et al. found that HIF-1α affects the expression of VEGF and MMP9 in H22 cells. Administration of saikosaponin b2, which has antipyretic and hepatoprotective effects, can inhibit the expression of HIF-1α, MMP2, MMP9, and VEGF in H22 cells and significantly weaken their angiogenic ability [85]. Wu et al. found that, after infecting HepG2 cells with Plasmodium, the expression of HIF-1α, MMP2, and MMP9 in HepG2 cells was decreased, and cell proliferation was blocked. They reached the same conclusion that HIF-1α can promote the progression of liver cancer by affecting MMP2 and MMP9 [86]. Ning et al. found that chitin inhibitors can promote the expression of HIF-1α in HepG2 cells and that HIF-1α can upregulate the expression of MMP2. Administration of HIF-1α inhibitors can block the upregulation of MMP2 and delay the progression of liver cancer, providing further verification of the conclusion that the HIF-1α/MMP2 or HIF-1α/MMP9 signaling pathways are involved in the occurrence and development of liver cancer [87]. Similarly to gastric cancer, inhibiting the upstream signaling pathways of HIF-1α in liver cancer cells can also effectively delay the proliferation of liver cancer cells. Zheng et al. established a tumor-bearing mouse model using SMMC-7721 cells and treated them with frankincense and myrrh. They found that frankincense and myrrh could significantly reduce the expression of EGFR, subsequently decreasing the expression of MAPK, HIF-1α, and ErbB. The tumor volume of the tumor-bearing mice decreased, and the concentration of multiple molecules in the peripheral blood of the mice, including HIF-1α, TNF-α, VEGF, and MMP-9, was reduced [88]. The reason for this is that frankincense and myrrh inhibited the phosphorylation process of multiple targets downstream of EGFR, causing the blockage of the PI3K/Akt and MAPK signaling pathways and subsequently inducing the downregulation of HIF-1α and MMPs.

Pancreatic cancer is the digestive system tumor with the poorest prognosis and survival time. HIF-1α plays an important role in the occurrence and development of pancreatic cancer. Chen et al. found that hypoxia could significantly increase the protein expression of HIF-1α in the pancreatic cancer cell line BxPC-3 and then induce the expression of MMP2 and MMP9. Treating BxPC-3 cells with the HIF-1α inhibitor KC7F2 could reduce their cell proliferation ability. Treating BxPC-3 cells with salidroside could also achieve similar effects. The mechanism is that salidroside inhibits the expression of HIF-1α, MMP2, and MMP9 in BxPC-3 cells, making it a potential adjuvant drug for the treatment of pancreatic cancer [89]. Other HIF-1α inhibitors can also block the expression of MMP2 and MMP9. Shan et al. used 2-methoxyestradiol to significantly inhibit the proliferation of pancreatic cancer in mouse models. Immunohistochemistry showed that the HIF-1α/MMP2 and HIF-1α/MMP9 signaling pathways in the tumor model were blocked, proving the role of the HIF-1α/MMPs signaling pathway in the occurrence and development of pancreatic cancer [90]. Moreover, inhibiting the upstream signals of HIF-1α in pancreatic cancer can also effectively reduce the expression of MMPs. Sun et al. reported that the expression of the Von Hippel-Lindau (VHL) gene was downregulated in pancreatic cancer tissues, and overexpression of the VHL gene could significantly inhibit the HIF-1α/MMP2 and HIF-1α/MMP9 signaling pathways and inhibit the growth rate of pancreatic cancer cells in mice [91].

Research on the HIF-1α/ADAM10 signaling pathway in tumor occurrence and development is relatively scarce. Ou et al. pointed out that miR-153 targets HIF-1α for degradation. miR-153 is lowly expressed in pancreatic cancer cells and is insufficient for the inhibition of the expression of HIF-1α, leading to increased HIF-1α expression and the induction of ADAM10 expression, resulting in an increase in sMICA and hindering the activation of NK cells, causing the internalization of NKG2D receptors and preventing NK cells from effectively controlling tumor metastasis [58]. The HIF-1α/ADAM10 signaling pathway has also been reported to promote the generation of sMICA in prostate cancer and hinder the killing effect of immune cells [29]. However, in general, the research on this signaling pathway in tumor immune escape is relatively limited.

In summary, the HIF-1α/MMPs signaling pathway and the HIF-1α/ADAMs signaling pathway are generally abnormally activated in digestive system tumors. MMPs or ADAMs can hydrolyze MICA and MICB on the surface of tumor cells into soluble molecules. Subsequently, sMICA and sMICB inhibit the function of NKG2D and damage the cytotoxic effect of NK cells, causing tumor cells to be unable to be effectively killed by NK cells and CD8^+^ T cells, mediating tumor immune escape and disease progression. The related mechanisms are shown in Figure 3 and Figure 4.

HIF-1α is generally highly expressed in digestive system tumors such as liver cancer, gastric cancer, colorectal cancer, pancreatic cancer, and gallbladder cancer. As a transcription factor, HIF-1α activates and promotes the expression of downstream metalloproteinases (including MMPs and ADAMs) and mediates the generation of soluble MICA and soluble MICB. The HIF-1α/MMPs and HIF-1α/ADAMs signaling pathways play a vital role in the immune escape of digestive system tumors. This diagram was drawn using Figdraw 2.0software (www.figdraw.com, accessed on 27 March 2025); the ID on the Figdraw website is AIYAW408e9.

MICA and MICB on the surface of tumor cell membranes can bind to the NKG2D receptor on the surface of NK cells, activating NK cells to perform cytotoxic effects. HIF-1α is highly expressed in various digestive system tumor cells and can promote the expression of downstream metalloproteinases (including MMPs and ADAMs). Metalloproteinases hydrolyze MICA and MICB on the surface of tumor cells to generate soluble MICA and MICB (sMICA and sMICB). These soluble molecules can also bind to the NKG2D receptor. However, they cannot activate NK cells, and the soluble ligands bind to NKG2D, which hinders the contact between NK cells and tumor cells, allowing tumor cells to evade the killing effect of NK cells. This diagram was drawn using Figdraw 2.0 software (www.figdraw.com, accessed on 27 March 2025); the ID on the Figdraw website is WIRIU5bff5.

## 6. Summary and Outlook

Malignant tumors of the digestive system have a serious incidence and poor treatment outcomes. Immune escape is a significant cause of tumor progression. The high expression of MICA and MICB on the surface of various epithelial cell-derived malignant tumors makes them potential therapeutic targets and important mediators of tumor immune escape. Multiple metalloproteinases hydrolyze MICA and MICB and convert them into soluble ligand molecules that inhibit NKG2D signal activation and impede the killing of tumor cells by NK cells and CD8^+^ T cells. Abnormal expression of metalloproteinases is regulated by HIF-1α, which induces the expression of MMP2, MMP9, ADAM10, etc., in tumor cells, hydrolyzing MICA and MICB on the surface of tumor cells and mediating immune escape.

Currently, there are many studies on the use of the MICA-NKG2D immune molecule to kill tumors. The NKG2D-IL-15 bifunctional fusion protein prepared by Chen al. is a typical biological agent for activating NK cells to kill tumor cells [63]. Other similar studies have also been reported. Wang et al. prepared a fusion protein composed of anti-CD24 single-chain antibody and the extracellular domain of MICA, which could guide NK cells to kill CD24-expressing liver cancer cells in vitro [92]. However, these fusion proteins also have limitations. For example, the NKG2D-IL-15 fusion protein needs to bind to MICA on the surface of tumor cells, and this binding hinders the binding of NKG2D on NK cells to MICA on tumor cells, which is unfavorable for the activation of NK cells. Furthermore, the change in the immunogenicity of the fusion protein is also an important question that must be considered. It may be recognized as a foreign antigen in animals or humans, then triggering a specific immune response and causing rapid degradation and loss of function. Liu et al. prepared a ternary fusion protein MS-Fc composed of the extracellular domain of MICA, anti-CD20 single-chain antibody, and the Fc segment of human IgG1. They found that MS-Fc could effectively induce NK cells to kill CD20-expressing B lymphoma cells and significantly prolong the half-life of MS-Fc in mice, and no immune rejection reaction against MS-Fc occurred, making it a promising biological treatment method for the future [93].

Since the abnormal activation of the HIF-1α/MMPs or HIF-1α/ADAMs signaling pathway leads to sMICA and sMICB, which hinder the killing by NK cells and cause tumor progression and worsening, inhibiting this pathological activation of these signaling pathways is crucial for enhancing the anti-tumor immunity and carrying out immunotherapy. So far, the US FDA has not approved any inhibitors of metalloproteinases, mainly because metalloproteinases have important physiological significance, and the failure to timely hydrolyze some molecules may trigger pathological reactions [94]. However, many inhibitors of HIF-1α have passed phase II clinical trials and achieved high remission rates in cancer patients with high HIF-1α expression [95,96].

In addition, inhibiting the shedding of NKG2D ligands from the surface of tumor cells is also an important way to enhance the cytotoxic effect of NK cells on tumors. The α1 and α2 domains of MICA and MICB are the binding domains with NKG2D receptor, while the α3 domain is the action site of metalloproteinases. The use of monoclonal antibodies against the MICA α3 domain to inhibit the generation of sMICA and sMICB has been proven to have the effect of enhancing the killing of NK cells [97]. These antibodies not only do not hinder the binding of MICA/B to NKG2D but also can activate NK cells through their Fc fragments and enable NK cells to secrete anti-tumor cytokines such as IFN-γ [98]. The newly emerged monoclonal antibody CLN-619 with a strong affinity with the MICA/B α3 domain and can effectively increase the expression of MICA/B on the cell surface while reducing the generation of sMICA/B. CLN-619 activates NK cells through Fc fragments and triggers the ADCC effect, effectively controlling tumor spread in experimental mice [99]. In another study, the high-affinity monoclonal antibody RDM028 targeting the α3 domains of MICA and MICB was also reported to have inhibitory effects on the exfoliation of MICA and MICB, enhance the anti-tumor activity of NK cells, and promote the interaction between MICA/B and NKG2D [100]. Although these studies failed to inhibit the generation of soluble NKG2D ligands from the perspective of metalloproteinases, they blocked the hydrolysis of metalloproteinases and, to a certain extent, reversed the immune escape of tumor cells against NK cells, making them potential candidate immune checkpoints for future anti-tumor immunotherapy.

According to the content summarized in this article, after HIF-1α is inhibited, the expression of multiple metalloproteinases is reduced, and the concentrations of sMICA and sMICB are decreased; NK cells in cancer patients are relieved from inhibition and resume their anti-tumor activation. The regulation of MICA and MICB by HIF-1α and metalloproteinases is of great significance for understanding tumor progression and molecular targeting. Reports on the HIF-1α/MMPs or HIF-1α/ADAMs signaling pathway in digestive system malignant tumors are still relatively rare. In addition, whether other metalloproteinases are also regulated by HIF-1α to mediate the immune escape of tumor cells from the killing by NK cells, and whether common metalloproteinases (MMP2, MMP9, and ADAM10) are also regulated by factors parallel to HIF-1α, still require further research. The molecular regulatory network targeting MICA-NKG2D may become a target for cancer treatment and may also open up new avenues for the next generation of immunotherapy.

## Figures and Tables

**Figure 1 biomolecules-15-00899-f001:**
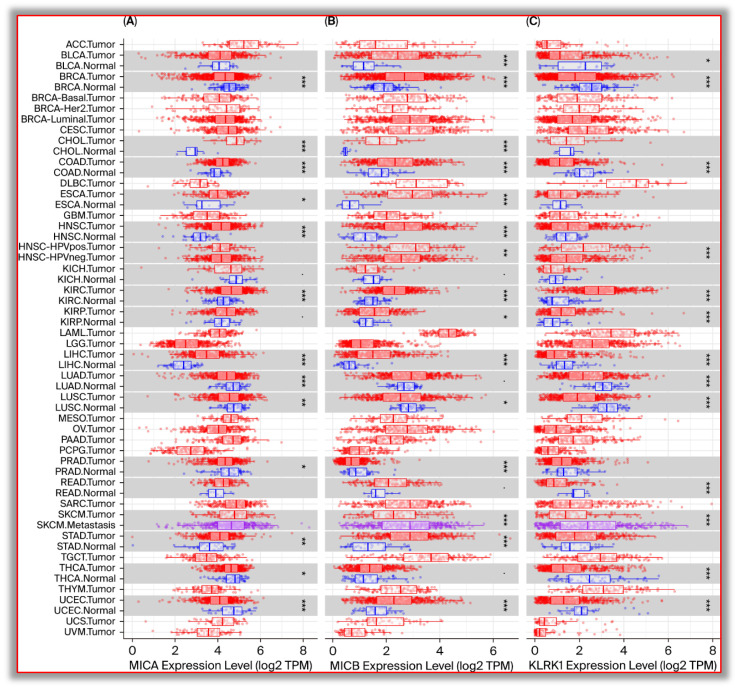
Expression of MICA, MICB, and NKG2D in cancers. (**A**). The expression levels of MICA in common tumors and their corresponding normal tissues; (**B**). The expression levels of MICB in common tumors and their corresponding normal tissues; (**C**). The expression levels of NKG2D in common tumors and their corresponding normal tissues. * *p* < 0.05, ** *p* < 0.01, *** *p* < 0.001.

**Figure 2 biomolecules-15-00899-f002:**
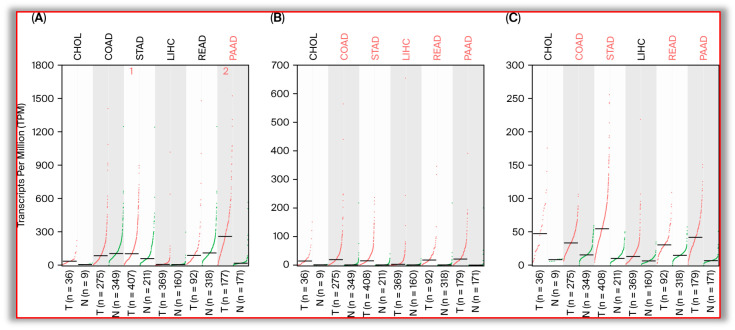
Expression levels of MMP2, MMP9, and ADAM10 in digestive system tumors.

**Figure 3 biomolecules-15-00899-f003:**
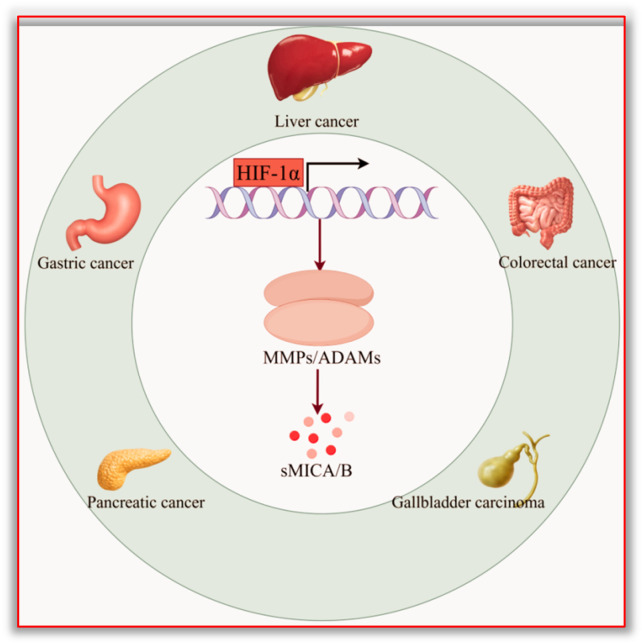
The generation of sMICA promoted by the HIF-1α/MMPs and HIF-1α/ADAMs signaling pathways.

**Figure 4 biomolecules-15-00899-f004:**
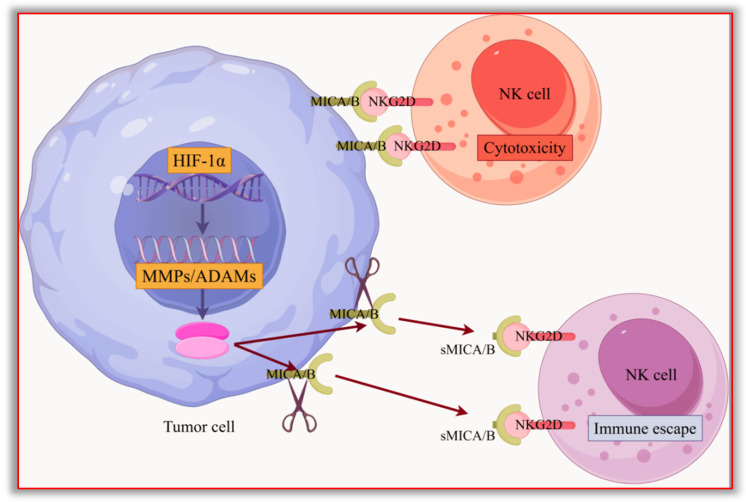
HIF-1α/MMPs and HIF-1α/ADAMs signaling pathways promote sMICA and sMICB to help tumor cells against the killing effect of NK cells.

**Table 1 biomolecules-15-00899-t001:** Statistics on the expression of HIF-1α in digestive system malignant tumors and its correlation with the survival time of patients.

Cancer Types	Expression Level of HIF-1α	Correlation of Survival Time
STAD [67]	high	negative
COAD [68]	high	negative
PAAD [69]	high	negative
LIHC [70]	high	negative
GBC [71]	high	negative
CHOL [72]	high	negative

Note: STAD: gastric cancer. COAD: colorectal cancer. PAAD: pancreatic cancer. LIHC: hepatocellular carcinoma. GBC: gallbladder cancer. CHOL: cholangiocarcinoma.

## Data Availability

Not applicable.

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
