# Peer review of "HIF-1α: A Key Factor Mediating Tumor Cells from Digestive System to Evade NK Cell Killing via Activating Metalloproteinases to Hydrolyze MICA/B"

_biomolecules, 2025, doi:10.3390/biom15060899_

Round 1

Reviewer 1 Report

Comments and Suggestions for Authors

The main question that is considered in the review.

It is known that MICA and MICB molecules are related to the molecules of the major complex of MHC class I and are expressed on the membranes of transformed cells. These glycoproteins bind to the NKG2D receptor of NK cells, which leads to the activation of their cytotoxic reaction in relation to cells expressing MICA and/or MICB. In turn, under the action of proteinases, soluble forms of MICA/B proteins are formed.

Binding of soluble forms of ligands (sMICA and sMICB) to NKG2D leads to a decrease in NK cell activity.

The growth of a number of tumors of the gastrointestinal tract, pancreas, liver, kidneys, lungs, skin and circulatory system is accompanied by an increase in the concentration of soluble forms of MICA/B in the blood plasma of patients.

The HIF-1α/MMP or HIF-1α/ADAM signaling pathway may be considered as key elements for NK cell immune evasion in gastrointestinal tumors.

This article reviews is devoted to the study of the molecular mechanisms of common digestive system malignancies evading NK cell killing, providing new insights into the mechanism of tumor immune escape.

Review structure.

The analysis of this review showed its high exclusivity. The review is well structured and contains sections devoted to the analysis of the level of MICA/B and NKG2D are imbalanced in expression in tumor tissues of the digestive system, abnormal activation of metalloproteinases leads to the shedding of MICA/B on tumor cell surface to form soluble ligands, soluble MICA/B impairs NKG2D signaling, and the key factor for digestive system tumors to evade killing by NK cells is that the HIF-1α promotes the expression of metalloproteinase. In conclusion, the authors offer two figures (Fig. 3, 4) summarizing the analyzed information. Although there is insufficient information about the HIF-1α/MMP or HIF-1α/ADAM signaling pathway in the actual malignant tumor system, the authors analyzedall available cases. The current limitations are discussed and it is concluded that the MICA-NKG2D-targeted molecular network may be a target for treatmentand may become a platform for novel immunotherapy.

There are no fundamental comments.

What is the value of this review?

Although there are related reviews (e.g. https://doi.org/10.3892/or.2024.8796; https://doi.org/10.1007/s12192-014-0532-5; https://jenci.springeropen.com/articles/10.1186/s43046-023-00186-z et ctr.), none of them covers all sections of this review. Namely, the review covers a clearly limited topic related to the analysis of HIF-1α as the main factor inducing the escape of gastrointestinal tumor cells from NK cells through the activation of MMZ3 and subsequent hydrolysis of MICA/B.

Figures&table

Figures 1 and 2 highlight the relevance of this review by summarizing the data on MICA/B and NKG2D expression in digestive system tumors and the expression level of MMP2, MMP9 and ADAM10 in digestive system tumor. The comments are provided below.

Figures 3 and 4 successfully summarize the information presented in the review. In particular, a scheme is presented showing the involvement of HIF-1a/MMPs and HIF-1a/ADAMs in the modification of MICA/B molecules leading to tumor cell escape from NK cells.

Table 1 summarizes the data on HIF-1a expression in the gastrointestinal tract and correlation with patient survival.

The references

The references provided in the review are appropriate, the first author cites his article in references 23 and 56, it is important to note that these are the same article.

[23] Luo Q, Luo W, Zhu Q, et al. Tumor-Derived Soluble MICA Obstructs the NKG2D Pathway to Restrain NK Cytotoxicity [J]. Aging

Dis. 2020, 11(1):118-128.

[56] Luo Q, Luo W, Zhu Q, et al. Tumor-Derived Soluble MICA Obstructs the NKG2D Pathway to Restrain NK Cytotoxicity [J]. Aging

Dis. 2020, 11(1):118-128.

Notes:

Figure 1. It might be better to indicate in a note under the figure the resource that was used (https://cistrome.shinyapps.io/timer/)

Figure 2. The inscriptions on the scales are difficult to read. It would be better to indicate which resource was used to construct the figure.

Line 135 is a typo with The capitalized.

References 23 and 56 are duplicates.

For authors

This article reviews is devoted to the study of the molecular mechanisms of common digestive system malignancies evading NK cell killing, providing new insights into the mechanism of tumor immune escape. The analysis of this review showed its high exclusivity. There are no fundamental comments.

Notes:

Figure 1. It might be better to indicate in a note under the figure the resource that was used (https://cistrome.shinyapps.io/timer/)

Figure 2. The inscriptions on the scales are difficult to read. It would be better to indicate which resource was used to construct the figure.

Line 135 is a typo with The capitalized.

The references provided in the review are appropriate, the first author cites his article in references 23 and 56, it is important to note that these are the same article.

[23] Luo Q, Luo W, Zhu Q, et al. Tumor-Derived Soluble MICA Obstructs the NKG2D Pathway to Restrain NK Cytotoxicity [J]. Aging

Dis. 2020, 11(1):118-128.

[56] Luo Q, Luo W, Zhu Q, et al. Tumor-Derived Soluble MICA Obstructs the NKG2D Pathway to Restrain NK Cytotoxicity [J]. Aging

Dis. 2020, 11(1):118-128.

Author Response

Dear Reviewers,

We appreciate the opportunity to revise our manuscript and are grateful for the insightful comments. We greatly appreciate the valuable comments provided, which are instrumental in revising and enhancing our manuscript, as well as guiding our research. 

Here, we have included detailed responses to each of your feedback. All the modifications in our manuscript have retained the traces of modification. To improve the figure appearance, we adjusted the resolution of figures, while keeping the data itself unchanged. In this cover letter, we provided point-to-point responses to your comments, and we deeply hope that the new manuscript fulfills the publication requirements.

Comment 1: Figure 1. It might be better to indicate in a note under the figure the resource that was used (https://cistrome.shinyapps.io/timer/)

Response 1: Thank you for your meticulous observation. We have supplemented the annotations to Figure 1, detailing the meaning we want to convey to the readers through this figure and attaching the source of the figure (https://cistrome.shinyapps.io/timer/). For details, please refer to lines 144-152 of our revised manuscript. Thank for your good suggestion.

Comment 2: Figure 2. The inscriptions on the scales are difficult to read. It would be better to indicate which resource was used to construct the figure.

Response 2: Thank you for your meticulous observation. We have redrawn the figure, enhancing its resolution and clarity while keeping the information in this figure unchanged. Furthermore, we have added detailed annotations to Figure 2 and pointed out the source of this figure simultaneously in the main text and the figure captions. For details, please refer to lines 185-198 of our revised manuscript. Thank you very much.

Comment 3: Line 135 is a typo with The capitalized.

Response 3: Thank you for your meticulous observation. We checked all the words in our manuscript and corrected the words with spelling mistakes.

Comment 4: The references provided in the review are appropriate, the first author cites his article in references 23 and 56, it is important to note that these are the same article.

Response 4: Thank you very much for your meticulous observation. We are very sorry about this. We have replaced the reference in Ref 23 and confirmed that the new reference does not duplicate the others. The new reference is “Kegasawa T, Tatsumi T, Yoshioka T, et al. Soluble UL16-binding protein 2 is associated with a poor prognosis in pancreatic cancer patients [J]. Biochem Biophys Res Commun. 2019, 517(1):84-88.” Since we have added some references this time, this reference is currently the 25th one. Thank you.

Reviewer 2 Report

Comments and Suggestions for Authors

Dear Authors

Your review covers an important topic and provides a comprehensive summary of how HIF-1α contributes to immune escape in digestive system tumors through metalloproteinase activation and MICA/B shedding. The content is well-researched and relevant.

Also, consider simplifying and condensing some dense sections to improve flow. A clearer discussion of future research directions and therapeutic implications would strengthen the conclusion.

Comments on the Quality of English Language

The English language needs improvement for clarity and readability. Please revise the grammar and sentence structure throughout the manuscript.

Author Response

Dear Reviewers,

We appreciate the opportunity to revise our manuscript and are grateful for the insightful comments. We greatly appreciate the valuable comments provided, which are instrumental in revising and enhancing our manuscript, as well as guiding our research. 

Here, we have included detailed responses to each of your feedback. All the modifications in our manuscript have retained the traces of modification. To improve the figure appearance, we adjusted the resolution of figures, while keeping the data itself unchanged. In this cover letter, we provided point-to-point responses to your comments, and we deeply hope that the new manuscript fulfills the publication requirements.

Comment 1: Consider simplifying and condensing some dense sections to improve flow. A clearer discussion of future research directions and therapeutic implications would strengthen the conclusion.

Response 1: Thanks for your good suggestion. In the part of discussion, we conducted an in-depth discussion on the current treatment methods for controlling the generation of soluble MICA/B, and added the content that anti-MICA/B α3 domain monoclonal antibodies enhance the anti-tumor effect of NK cells. This is the emerging immunotherapy method targeting the MICA/B-NKG2D axis at present. The newly added content is as follows, which is in the penultimate paragraph of the discussion in our revised manuscript. Thanks very much.

In addition, inhibiting the shedding of NKG2D ligands from the surface of tumor cells is also an important way to enhance the cytotoxic effect of NK cells on tumors. The α1 and α2 domains of MICA and MICB are the binding domains with NKG2D receptor, while the α3 domain is the action site of metalloproteinases. The use of monoclonal antibodies against the MICA α3 domain to inhibit the generation of sMICA and sMICB has been proven to have the effect of enhancing the killing of NK cells [95]. These antibodies not only do not hinder the binding of MICA/B to NKG2D, but also can activate NK cells through their Fc fragments and enable NK cells to secrete anti-tumor cytokines such as IFN-γ [96]. The newly emerged monoclonal antibody CLN-619 with a strong affinity with the MICA/B α3 domain and can effectively increase the expression of MICA/B on the cell surface, while reducing the generation of sMICA/B. CLN-619 activates NK cells through Fc fragments and triggers the ADCC effect, effectively controlling tumor spread in experimental mice [97]. In another study, the high-affinity monoclonal antibody RDM028 targeting the α3 domains of MICA and MICB was also reported to have inhibitory effects on the exfoliation of MICA and MICB, enhance the anti-tumor activity of NK cells, and promote the interaction between MICA/B and NKG2D. [98] Although these studies failed to inhibit the generation of soluble NKG2D ligands from the perspective of metalloproteinases, they blocked the hydrolysis of metalloproteinases and to a certain extent reversed the immune escape of tumor cells against NK cells, making them potential candidate immune checkpoints for future anti-tumor immunotherapy.

Comment 2: The English language needs improvement for clarity and readability. Please revise the grammar and sentence structure throughout the manuscript.

Response 2: Thank you for your good suggestion. We have invited a professional language service team to improve the grammar of our manuscript and uploaded the English editing certificate at the same time, hoping it will meet the requirements of the journal. Thank you very much.

Reviewer 3 Report

Comments and Suggestions for Authors

The review entitled „HIF-1α: A key factor mediating tumor cells to evade NK cell killing by activating metalloproteinases to hydrolyze MICA/B” is focused on presenting the HIF-1α/MMPs or HIF-1α/ADAMs signalling pathways and immune escape mechanisms in digestive system tumors. This manuscript is well-organised and detailed, with a logical flow. However, I have several minor comments to be addressed:

  1. Figures 1 and 2 are unclear and require enhancement.
  2. Lack of references in Figures 1. What was the source of the data? Authors describe results in a paper without providing references.
  3. Figure 2: The author relies on the findings of others without recognising their origin. What was the source of the data?
  4. The Figures reflect the text information. However, the Figure legends must provide the reader with sufficient information to understand the figure without relying on the main text.
  5. Figure 1 shows not only digestive system tumors.
  6. The lack of references in many parts of the article:
  • Lines 132-140: reference should be added
  • Lines 183-190: Some reference is needed.
  • Lines 330-339, 345-350: Some reference is needed.
  1. Lines 261-265: The font should be reduced.

Author Response

Dear Reviewers,

We appreciate the opportunity to revise our manuscript and are grateful for the insightful comments. We greatly appreciate the valuable comments provided, which are instrumental in revising and enhancing our manuscript, as well as guiding our research. 

Here, we have included detailed responses to each of your feedback. All the modifications in our manuscript have retained the traces of modification. To improve the figure appearance, we adjusted the resolution of figures, while keeping the data itself unchanged. In this cover letter, we provided point-to-point responses to your comments, and we deeply hope that the new manuscript fulfills the publication requirements.

Comment 1: Figures 1 and 2 are unclear and require enhancement.

Response 1: Thank for your good suggestion. We have redrawn the figure, enhancing its resolution and clarity while keeping the information in this figure unchanged. For details, please refer to these two figure in page 5 and 6. Thank you very much.

Comment 2: Lack of references in Figures 1. What was the source of the data? Authors describe results in a paper without providing references. 

Response 2: Thank you for your meticulous observation. We added detailed annotations to this figure and provided the sources of the data in this figure. For details, please refer to the page 5. Thank you very much.

Comment 3: Figure 2: The author relies on the findings of others without recognising their origin. What was the source of the data? 

Response 3: Thank you for your meticulous observation. Like the Figure 1, we also added detailed annotations and provided the sources of the data in this figure. For details, please refer to the page 6. Thank you very much.

Comment 4: The Figures reflect the text information. However, the Figure legends must provide the reader with sufficient information to understand the figure without relying on the main text. 

Response 4: Thank for your good suggestion. We have made detailed annotations for all the figures in our manuscript, including Figures 1 to 4. Furthermore, we have also annotated the abbreviations for the table 1. For details, please refer to the page 5, 6, 10, 14 and 15. Thank you very much.

Comment 5: Figure 1 shows not only digestive system tumors 

Response 5: Thank you for your kindly suggestion. Since this figure is sourced from a public database and is a file showing the expression levels of specific genes in all common tumors, it is indeed impossible to screen for specific tumor types. Based on this, we have changed the figure title to “Expression of MICA, MICB and NKG2D in cancers” to avoid causing misunderstandings among readers. For details, please refer to the page 5. Thank you very much.

Comment 6: The lack of references in many parts of the article:

Lines 132-140: reference should be added

Lines 183-190: Some reference is needed.

Lines 330-339, 345-350: Some reference is needed.

Lines 261-265: The font should be reduced.

Response 6: Thank for your good suggestion. These contents do require supplementary references. We conducted a comprehensive review of the full text to ensure that all the cited content has references. However, the content in lines 132-140 (lines 135-141 in our newly revised manuscript) is the meanings of Figure 1 and not from other reference. Therefore, we did not add any references, but we have stated the sources of these data at the end of this paragraph. Thank you very much for your valuable guidance.